# Effect of Primary Crystals on Pore Morphology during Semi-Solid Foaming of A2024 Alloys

**Takashi Kuwahara [1,*], Mizuki Saito [1], Taro Osaka [1] and Shinsuke Suzuki [1,2]**

[1]   Faculty of Science and Engineering, Waseda University, 3-4-1 Okubo, Shinjuku, Tokyo 169-8555, Japan; m.saitoh@akane.waseda.jp (M.S.); t-evolva@moegi.waseda.jp (T.O.); suzuki-s@waseda.jp (S.S.)

[2]   Kagami Memorial Research Institute of Materials Science and Technology, Waseda University, 2-8-26 Nishi-Waseda, Shinjuku, Tokyo 169-0051, Japan

*   Correspondence: takuwahara@ruri.waseda.jp; Tel.: +81-(0)3-5286-8126

**Abstract:** We investigated pore formation in aluminum foams by controlling primary crystal morphology using three master alloys. The first one was a direct chill cast A2024 (Al-Cu-Mg) alloy (DC-cast alloy). The others were A2024 alloys prepared to possess fine spherical primary crystals. The second alloy was made by applying compressive strain through a Strain-Induced Melt-Activated process alloy (SIMA alloy). The third one was a slope-cast A2024 alloy (slope-cast alloy). Each alloy was heated to either 635 °C (fraction of solid $f_s$ = 20%) or 630 °C ($f_s$ = 40%). TiH$_2$ powder was added to the alloys as a foaming agent upon heating them to a semi-solid state and they were stirred while being held in the furnace. Subsequently, A2024 alloy foams were obtained via water-cooling. The primary crystals of the DC-cast alloy were coarse and irregular before foaming. After foaming, the size of the primary crystals remained irregular, but also became spherical. The SIMA and slope-cast alloys possessed fine spherical primary crystals before and after foaming. In addition to average-sized pores (macro-pores), small pores were observed inside the cell walls (micro-pores) of each alloy. The formation of macro-pores did not depend on the formation of the primary crystals. Only in the DC-cast alloy did fine micro-pores exist within the primary crystals. The number of micro-pores in the SIMA and slope-cast alloys was one third of that in the DC-cast alloy.

**Keywords:** semi-solid; aluminum foam; primary crystals; SIMA process; slope casting; pore morphology

## 1. Introduction

New, lighter materials are required to reduce the environmental impact and cost of transportation equipment, including automobiles and aircrafts [1,2]. In addition, passenger safety should be maintained or enhanced by these new materials. Aluminum foam has attracted significant attention in recent years for meeting weight reduction and safety requirements. Aluminum foam has many pores and, owing to its ultralight and excellent shock-absorbing properties, could be applied to transportation equipment. However, the compressive properties of aluminum foam must be improved before it can be effectively used for this purpose. The compressive properties of foams are determined by their base metal and structure [3]. To this end, Fukui et al. [4,5] improved the compressive properties of aluminum foam using the super duralumin A2024 (Al-Cu-Mg) alloy, which is a well-known high-strength and lightweight material, as a base metal to improve the strength of the foam.

Foaming via the melt route is a common and efficient fabrication method for aluminum alloy foams [6]. With this method, the melt must be thickened to maintain pore stability. In general, thickening is achieved by adding Ca, which generates oxides that spread throughout the melt [7]. To fabricate the A2024 alloy foams, the oxide of the alloying element, Mg, is used as a thickener instead

of Ca. However, decreases in base metal strength have been observed because the quantity of Mg as an alloying element also decreases.

To solve this problem, Hanafusa et al. [8] fabricated foams in a semi-solid state using primary crystals, instead of oxides, as a thickener. Sekido et al. [9] investigated the thickening effect of primary crystals in a semi-solid state. Our group demonstrated that it is possible to fabricate an A2024 alloy foam without adding a thickener in a semi-solid state. However, the effects of this fabrication on the properties of foam have been researched only experimentally, not systematically.

Also, the primary crystals acting as a thickener in the melt are larger in size than those of the oxides. Moreover, the primary crystals are generally larger than the preferred range of thickener sizes in the melt route [10]. Therefore, foams fabricated in a semi-solid state may have different stabilization mechanisms than those in the melt route. How these large thickeners affect the formation of pores in foams is not evident. However, compared to oxides, the sizes and shapes of primary crystals are easy to control and evaluate. Therefore, it may be possible to improve the foam fabrication process by aggressively controlling the primary crystals.

The objective of this study is to investigate the effects of the large primary crystals present inside the melt on pore formation in aluminum alloy foams in a semi-solid state. The effects of the diameter and shape of the crystals on pore morphology were examined. To observe the changes caused by the differences in the morphology of primary crystals, A2024 alloy foams were fabricated via the processing of an A2024 master alloy by controlling the formation of primary crystals to favor a fine spherical shape. The fabrication processes used were Strain-Induced Melt Activation (SIMA) [11] and slope casting [12]. Subsequently, we evaluated pore formation in the foam and the formation of primary crystals in the cell wall. In addition, we measured the Ti distribution to trace the path of $TiH_2$, which was the blowing agent used.

The innovations of this study are as follows. First, to systematically derive the effects of diameter, shape, and fraction of solid on primary crystals, the size and shape of the primary crystals were controlled by SIMA and slope casting. These controlling processes were difficult in oxide particle thickening. Second, we investigated fabrication through the stabilization of cell walls by controlled primary crystals.

## 2. Materials and Methods

### 2.1. Fabrication of Master Alloys

We fabricated three kinds of A2024 master alloys. Table 1 shows the composition of the A2024 alloys used in this study. Figure 1 shows a schematic of the fabrication method of the A2024 master alloys. The A2024 direct chill slab was sliced perpendicular to the pull-out direction of the DC casting. The A2024 plane was cut into several pieces. These pieces will be hereafter referred to as the DC-cast alloys. Then, some of the DC-cast alloys were processed via SIMA. The DC-cast alloys initially measured $20 \times 30 \times 30$ mm$^3$ and were compressed to a compressive strain $f_s = 10\%$, as done in the study presented by Sirong et al. [13]. Then, the pieces were annealed at 350 °C for 3 h in a muffle furnace. This master alloy will be hereafter referred to as the SIMA alloy. In addition, some of the DC-cast alloys were processed via slope casting. The A2024 pieces were heated at 649 °C and poured onto a cooling plate made of Cu. The degree of the slope, $\theta$, and the contact length, $l$, were 60° and 200 mm, respectively, as in a previous study [14]. Afterwards, the alloy was cast; this master alloy will be referred to as the slope-cast alloy.

**Table 1.** Chemical composition of the A2024 alloys.

| Element | Cu | Mg | Mn | Fe | Si | Cr | Zn | Ti | Al |
|---------|------|------|------|------|------|-------|-------|-------|------|
| mass% | 4.44 | 1.60 | 0.67 | 0.15 | 0.08 | <0.01 | <0.01 | <0.01 | bal. |

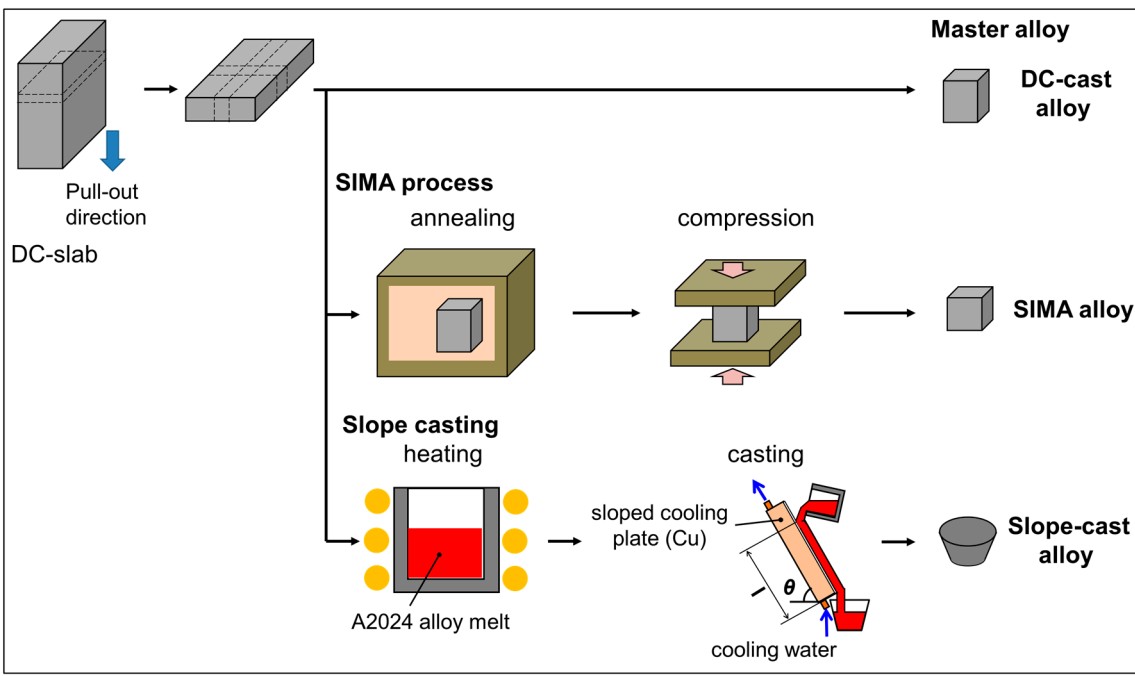

**Figure 1.** Schematic of the preparation of the master alloys.

### 2.2. Fabrication of A2024 Alloy Foam in a Semi-Solid State

Figure 2 shows the fabrication method of the A2024 alloy foam in a semi-solid state. First, 100 g of the DC-cast alloy, SIMA alloy, or slope-cast alloy was placed in a SUS304 crucible coated with $Al_2O_3$. Each alloy was heated to and held at 635 °C ± 0.3 °C (fraction of solid $f_s$ = 20%) [15] or 630 ± 0.3 °C ($f_s$ = 40%) [15] in an electric furnace, as shown in Figure 2a. The measured temperature was inside the error range. The temperature was measured using a K-type thermocouple covered with an $Al_2O_3$ protective tube. After the temperature stabilized, 1 mass% $TiH_2$ was added to the slurry after the thermocouple was taken out. Then, the slurry was stirred using an impeller coated with boron nitride at 900 rpm for 100 s, as shown in Figure 2b. After stirring, the slurry was held for 200 s and the crucible was removed from the furnace, as shown in Figure 2c. Subsequently, the foam was solidified and cooled using 3 L/min of water as shown in Figure 2d, and the A2024 alloy foam was obtained. Six foams were obtained in total, one for each combination of master alloy and fraction of solid. Additionally, to analyze the differences in the primary crystals before stirring and after foaming, metal samples held in a semi-solid state were obtained for each foam.

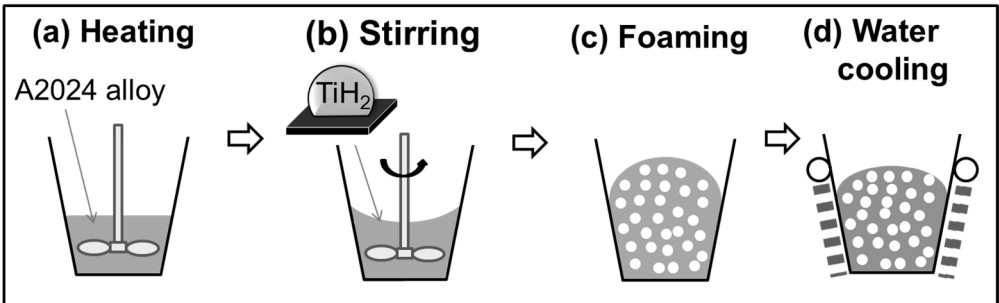

**Figure 2.** Fabrication method of the A2024 foam in a semi-solid state; (**a**) heating; (**b**) stirring; (**c**) foaming; (**d**) water cooling.

### 2.3. Evaluation of Pores and Primary Crystal Formation

To derive the porosity of the foam, we measured the volume of the foam via Archimedes' method using the mass of the foam and a spring scale. From Equation (1), we calculated the porosity *p* of the

foam using the density of the A2024 alloy foam $\rho_P$, which in turn was calculated using the volume and mass of the foams and the density of the A2024 alloy, $\rho_{NP} = 2.77$ Mg/m$^3$. The pore morphology (average pore diameter $d_p$ and average pore circularity $e_p$) of a cross-section from the middle of a sample was measured using the image-analysis software WinROOF$^{TM}$ version 6.1 (Mitani Corporation, Fukui, Japan). Equations (2) and (3) describe the pore formations, where $S$ is the area of the pores and $L$ is their perimeter. To improve the accuracy of our measurements, pores smaller than $d = 0.2$ mm were not considered. The diameter of the primary crystals $d_\alpha$ and their circularity $e_\alpha$ before and after the foaming of each alloy were also measured using WinROOF$^{TM}$ and calculated via Equations (2) and (3). Every measurement was taken once for every foam.

$$p = (1 - \rho_p/\rho_{NP}) \times 100\% \tag{1}$$

$$d_p = (4S/\pi)^{1/2} \tag{2}$$

$$e_\alpha = 4\pi S/L^2 \tag{3}$$

*2.4. Analysis of the Pure Ti Distribution*

We fabricated non-porous metal samples to estimate the distribution of the foaming agent and the foaming sites. These non-porous metal samples were fabricated using a similar procedure to the fabrication method for the A2024 foams, except that pure Ti was used instead of TiH$_2$. A Ti distribution analysis was performed via qualitative mapping using an electron probe micro-analyzer (EPMA, JEOL, JXA-8230, Tokyo, Japan; measurement elements: Ti, Cu; pressurization voltage: 15 kV; irradiation current value: $3 \times 10^{-8}$ A; irradiation interval: 10 μm; acquisition time: 20 ms/point; measurement range: 5 mm × 3.75 mm). Next, mapping images of Ti and Cu were processed using the following method. First, the Ti mapping image was made monochrome and binarized using WinROOF$^{TM}$ and Ti was colored green. Then, the parts that were not Ti were made transparent using the processing software paint.net Version 4.0.6. (dotPDN LLC, Washington, WA, USA). Finally, the Ti image and the monochrome Cu image were merged. One merged image was obtained for each DC-cast alloy and the SIMA alloy with a fraction of solid $f_s = 40\%$.

## 3. Results and Discussion

*3.1. Pore Formation on the A2024 Alloy Foams*

Figure 3 shows the microstructures of the (a) DC-cast, (b) SIMA, and (c) slope-cast A2024 alloys at $f_s = 40\%$. Figure 4 shows the average values of (a) the primary crystal diameter $d_\alpha$ and (b) the circularity $e_\alpha$ of the A2024 alloy foams. For Figure 4, approximately 100 primary crystals were measured for each sample, except for the after-foaming data of the DC-cast alloy, for which 27 primary crystals were measured. Dendritic primary crystals were present before melting and during holding at a semi-solid state in the DC-cast alloy, as shown in Figure 3a. Because some dendritic primary crystals grow while others do not, the diameters $d_\alpha$ of the primary crystals in the DC-cast alloy have high error values. In the SIMA alloy, shown in Figure 3b, dendritic primary crystals were present before melting, while fine and spherical primary crystals were formed during holding at a semi-solid state. In the slope-cast alloy, shown in Figure 3c, fine and spherical primary crystals were present before melting and during holding at a semi-solid state. Figures 3a and 4 show that the primary crystals were coarse and uneven in the DC-cast alloy before forming and remained coarse but spherical afterwards. The primary crystals in the SIMA and slope-cast alloys maintained their fine and spherical shape, as shown in Figure 3b,c and Figure 4. Therefore, the primary crystals of the DC-cast alloy spheroidized after foaming, whereas those of the SIMA and slope-cast alloys maintained their fine and spherical shape.

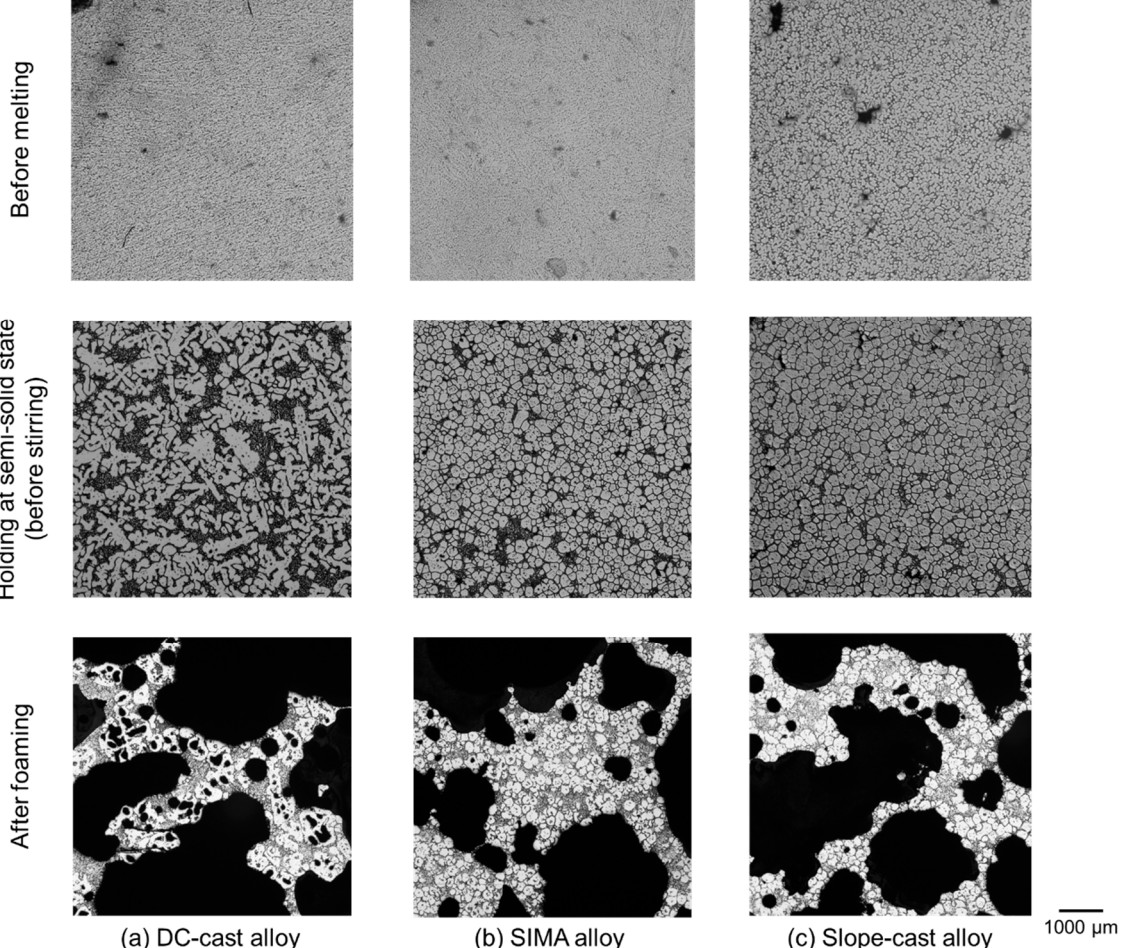

**Figure 3.** Images of the A2024 alloys ($f_s$ = 40%); (**a**) direct chill cast (DC-cast); (**b**) Strain-Induced Melt-Activated (SIMA); and (**c**) slope-cast.

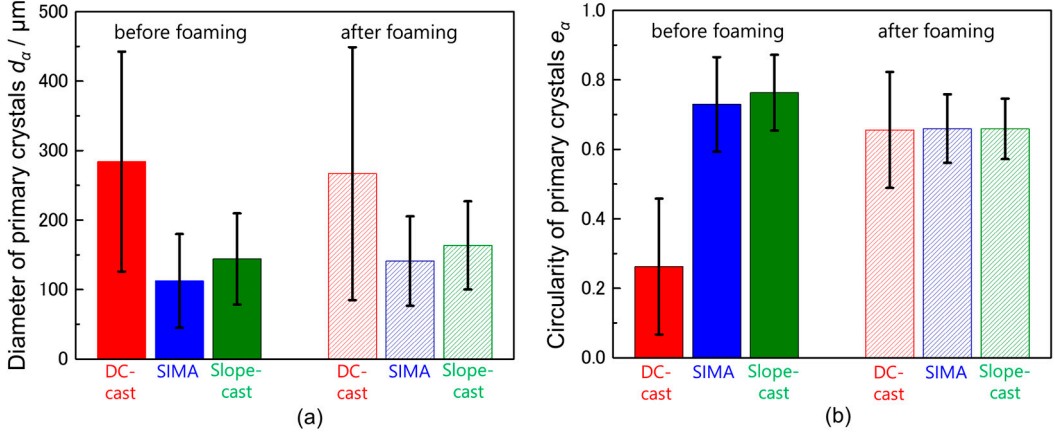

**Figure 4.** Average values of the primary crystals of the A2024 alloy foams fabricated via foaming in a semi-solid slurry ($f_s$ = 40%); (**a**) diameter $d_\alpha$; (**b**) circularity $e_\alpha$. The error bars show the standard deviations.

### 3.2. Macro-Pores of the A2024 Alloys Fabricated in a Semi-Solid State

Figure 5 shows cross-sections of the A2024 alloy foams of the DC-cast, SIMA, and slope-cast alloys with an $f_S$ of 20% and 40%. The pores were distributed homogeneously as shown in Figure 5. Although a non-porous part is conventionally found in the bottom part [4], the foams in this study did not show a clear difference of pore distribution at different heights. Therefore, the A2024 alloy foams of each alloy can be fabricated at $f_s$ values of 20% and 40%. All of the foams except for the slope-cast alloy foam fabricated at $f_s$ = 40% reached a porosity of 60%. Figure 6 shows the average pore (a) diameter $d_p$ and (b) circularity $e_p$ of the A2024 alloy foams with porosities of approximately 60%. In Figure 6, 1000–1900 pores were measured for each sample. The pores of the alloy foams fabricated at $f_s$ = 20% were finer and more spherical than those of the foams fabricated at $f_s$ = 40% (Figure 5; Figure 6). However, the differences between the primary crystals of the DC-cast and SIMA alloys had little effect on macro-pore morphology. Although the error value of the primary crystal size of the DC-cast was higher than that of the SIMA alloys, this difference did not affect macro-pore morphology. This result is due to the difference between the sizes of the primary crystals and the macro-pores. In each alloy, most primary crystals had a diameter $d_\alpha$ of less than 300 μm, whereas the diameter of the macro-pores $d_p$ was larger than 1000 μm. Therefore, differences in the formation of finer primary crystals had little effect on the formation of macro-pores in terms of macro-pore diameter. Moreover, as shown in Figure 6, the diameter of the macro-pores $d_p$ for a fraction of solid of 40% had high error values. These error values occur because a high fraction of solid makes it difficult for $TiH_2$ to be distributed uniformly. A schematic drawing of this mechanism is shown in Figure 7.

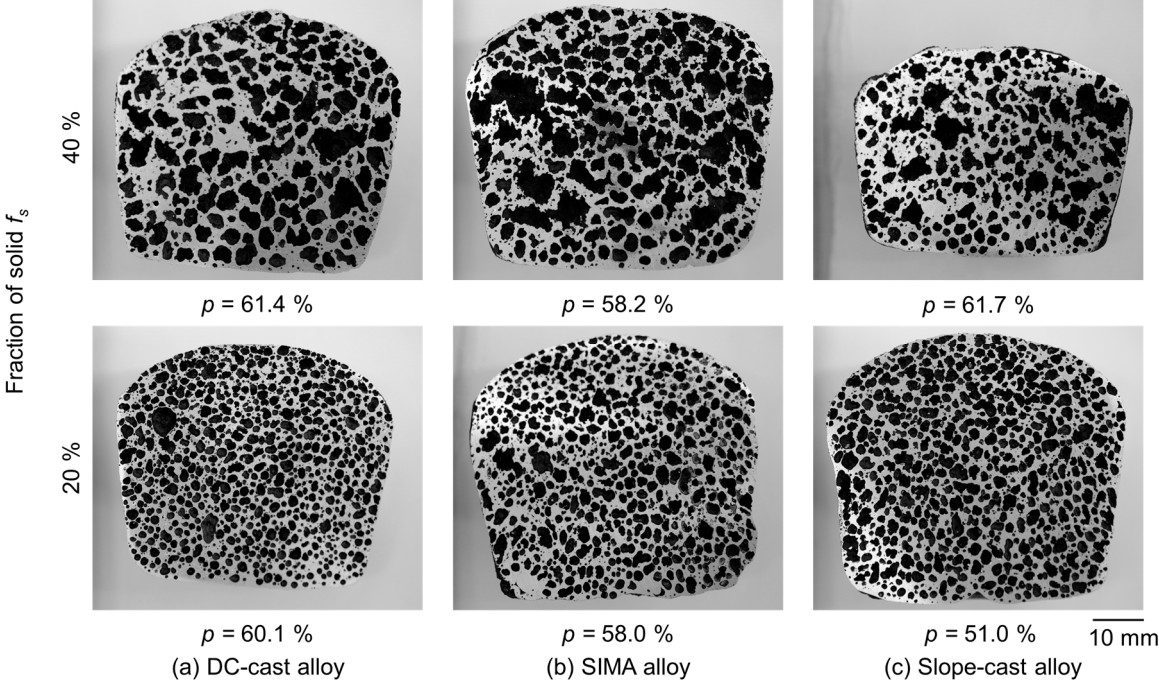

**Figure 5.** Cross-sections of foams: (**a**) direct chill cast (DC-cast); (**b**) Strain-Induced Melt-Activated (SIMA); and (**c**) slope-cast. *p*: porosity.

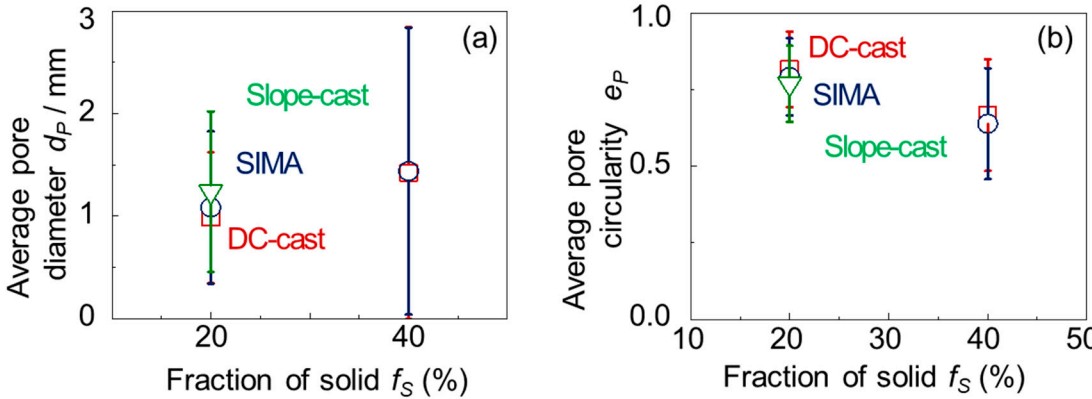

**Figure 6.** Average values of the pores of the A2024 alloy foams with a porosity of approximately 60%, fabricated via foaming in a semi-solid slurry: (**a**) pore diameter $d_p$, and (**b**) pore circularity $e_p$. The error bars show the standard deviations. DC-cast: direct chill cast, SIMA: Strain-Induced Melt-Activated.

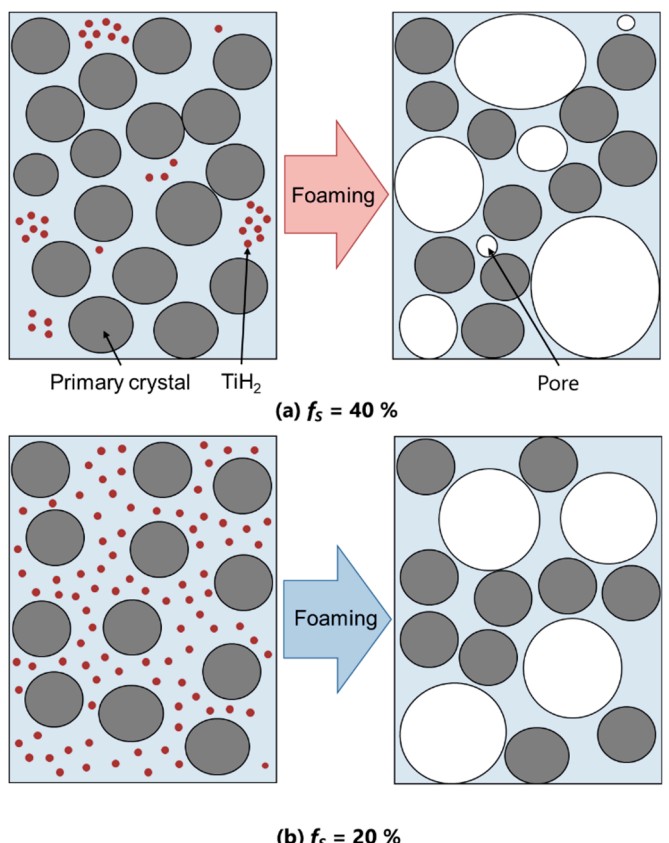

**Figure 7.** Schematic drawing of the $TiH_2$ distributed in the melt and pores after foaming. (**a**) $f_s = 40\%$; (**b**) $f_s = 20\%$.

### 3.3. Micro-Pores of the A2024 Alloys Fabricated in a Semi-Solid State

Figure 8 shows the location of the micro-pores in the A2024 alloy foams ($f_s = 40\%$) for the (a) DC-cast, (b) SIMA, and (c) slope-cast alloys. Figure 9 shows the number of micro-pores per unit area in all the A2024 alloy foams. The $f_S$ values of the foams were 20% and 40%. In this study, micro-pores are defined as pores in the cell wall with a diameter of less than 1000 μm, which is approximately the thickness of the cell walls. In addition, to improve image processing, pores with a diameter of less than 40 μm were ignored. Mukherjee et al. defined micro-pores as pores less than 350 μm in diameter [16]. Moreover, Ohgaki et al. reported that cracks initiate from pores with a diameter of 30 μm to 350 μm

when an aluminum foam is compressed [17]. However, as shown in Figure 10, most of the micro-pores in this study were inside this range. Therefore, the micro-pores measured in this study corresponded to those in previous studies. In the DC-cast alloy, micro-pores were present inside the primary crystals, as shown in Figure 8a, whereas micro-pores were present between the primary crystals in the SIMA and slope-cast alloys, as shown in Figure 8b. As indicated in Figure 9, the number of micro-pores in the SIMA and slope-cast alloys was approximately 1/3 of that in the DC-cast alloy. As the number of micro-pores increased, the strength of the base metal decreased. However, micro-pores also suppressed densification under compression, as reported by Toda et al. [18].

In addition, Figure 5, Figure 6, Figure 9, and Figure 10 show that there was no major difference in the results of each master alloy except for the number of micro-pores. In contrast, the fraction of solid seems to be the major factor that determines each property in aluminum foam. Furthermore, aluminum foam was fabricated three times for each fraction of solid. Because this foam seems to be reproducible, it was considered unnecessary to repeat the experiment.

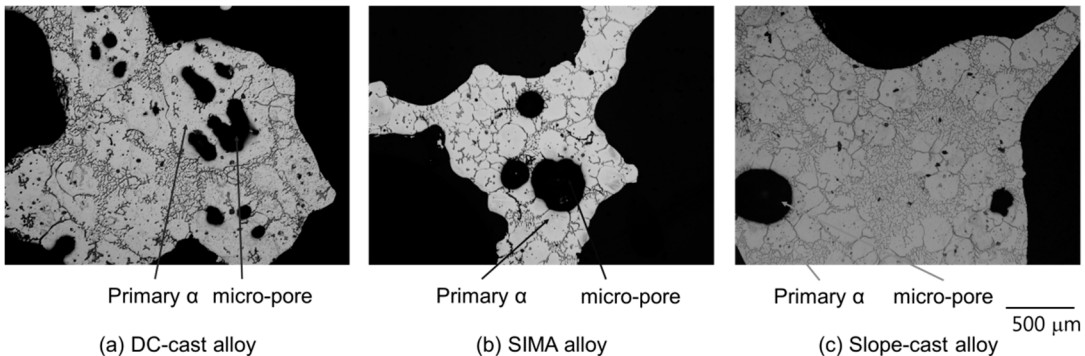

Primary α   micro-pore    Primary α        micro-pore    Primary α     micro-pore ______
                                                                                  500 μm
(a) DC-cast alloy              (b) SIMA alloy                  (c) Slope-cast alloy

**Figure 8.** Locations of micro-pores ($f_s$ = 40%). (**a**) direct chill cast (DC-cast) alloy; (**b**) Strain-Induced Melt-Activated (SIMA) alloy; and (**c**) slope-cast alloy.

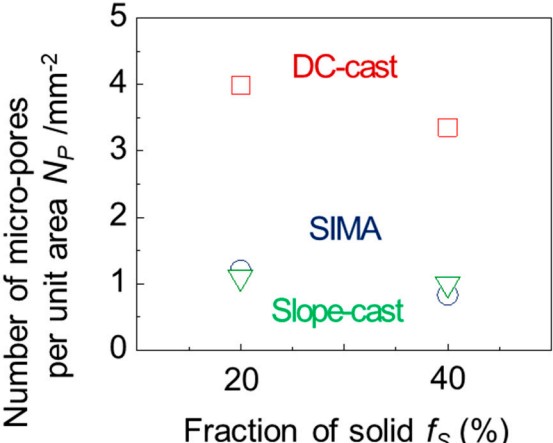

**Figure 9.** Number of micro-pores per unit area, $N_p$, in the A2024 alloy foams for the direct chill cast (DC-cast), Strain-Induced Melt-Activated (SIMA), and slope-cast alloys.

*3.4. Effect of Primary Crystals on Micro-Pore Formation*

As described in Section 3.2., the primary crystals were fine and spherical before and after stirring the SIMA and slope-cast alloys. Figure 11 shows the mappings for Ti and Cu obtained with the EPMA for the non-porous metal samples described in Section 2.4. In Figure 11, the darker parts with less Cu are considered to be primary crystals, which are mostly composed of Al. Moreover, Ti is not completely dispersed because the primary crystals exist as solids. For a better observation, Figure 12 shows the merged image made from the Ti and Cu images shown in Figure 11, as mentioned in Section 2.4. In the DC-cast alloy, the eutectic structure and Ti particles co-exist in one primary crystal, as shown in

Figure 12a. This eutectic structure was considered to have been liquid before solidification. In contrast, the liquid phase and the Ti particle exist between the primary crystals in the SIMA alloy, as shown in Figure 12b. From these results, we deduced a reason for the high quantity of micro-pores inside the DC-cast alloy. Figure 13 shows a schematic drawing of the mechanism behind micro-pore formation. As explained in Section 3.1, the uneven primary crystals in the DC-cast alloy spheroidize after foaming. Based on a previous study by Chen et al. [19], primary crystals have been shown to bend via stirring and spheroidize. Therefore, the liquid phase and $TiH_2$ were likely trapped in spheroidized primary crystals, similar to pure Ti, as shown in Figure 12a; for this reason, micro-pores could be found inside the primary crystals in the DC-cast alloy, as shown in Figure 13a.

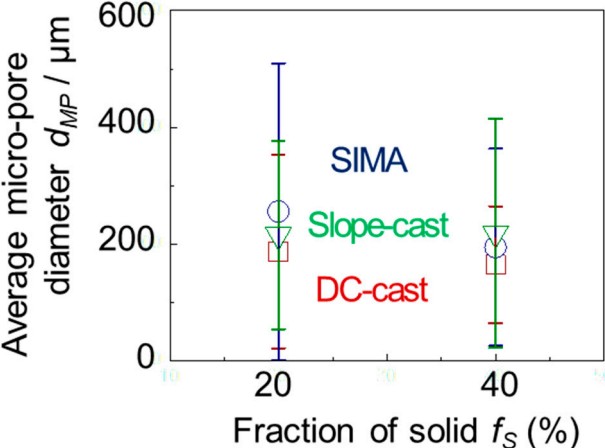

**Figure 10.** Average diameter of the micro-pores of the A2024 alloy foams fabricated via foaming in a semi-solid slurry for the direct chill cast (DC-cast), Strain-Induced Melt-Activated (SIMA), and slope-cast alloys.

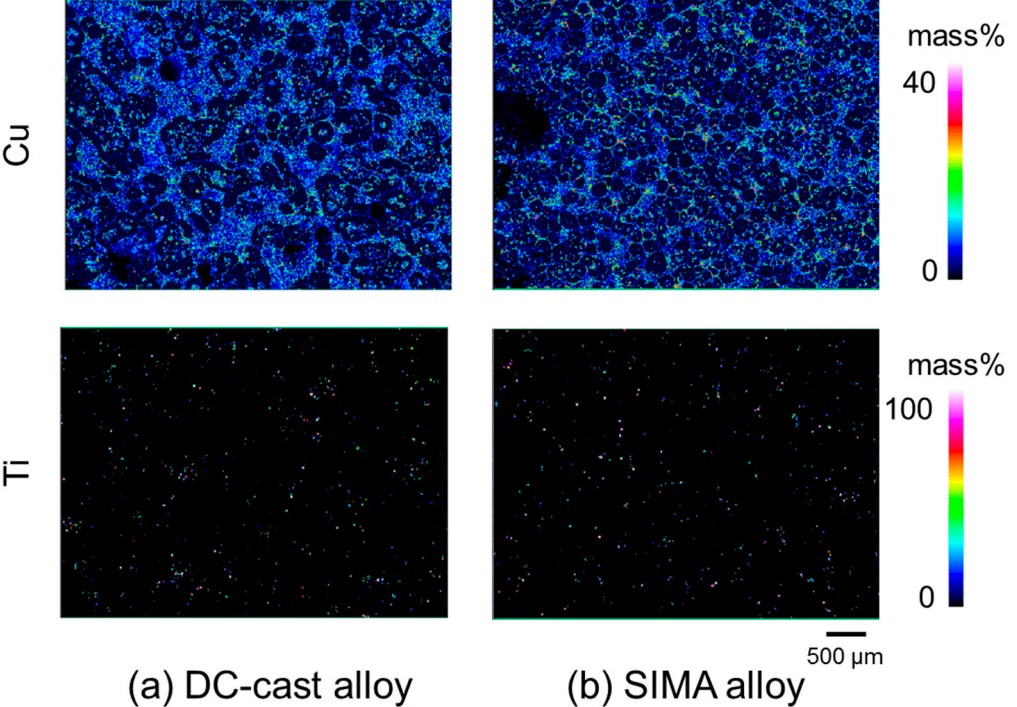

**Figure 11.** Mapping images of Ti and Cu obtained using the electron probe micro-analyzer (EPMA) ($f_s = 40\%$). (**a**) direct chill cast (DC-cast) alloy; (**b**) Strain-Induced Melt-Activated (SIMA) alloy.

However, when the primary crystals were spherical before foaming, TiH$_2$ rarely got trapped in the primary crystals while stirring, similar to pure Ti as shown in Figure 12b; this would happen in the SIMA and slope-cast alloys. Therefore, micro-pores were present between the primary crystals, as shown in Figure 13b. In addition, micro-pores between the primary crystals easily coalesce with other micro- or macro-pores because they are not completely wrapped by primary crystals. Consequently, the number of micro-pores in the SIMA and slope-cast alloys was 1/3 of that in the DC-cast alloy.

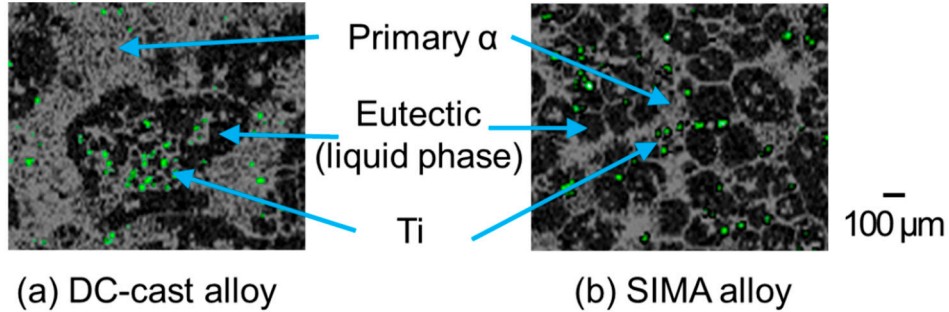

**Figure 12.** Merged images of Ti and Cu obtained using the electron probe micro-analyzer (EPMA) ($f_s$ = 40%). (**a**) direct chill cast (DC-cast) alloy; (**b**) Strain-Induced Melt-Activated (SIMA) alloy.

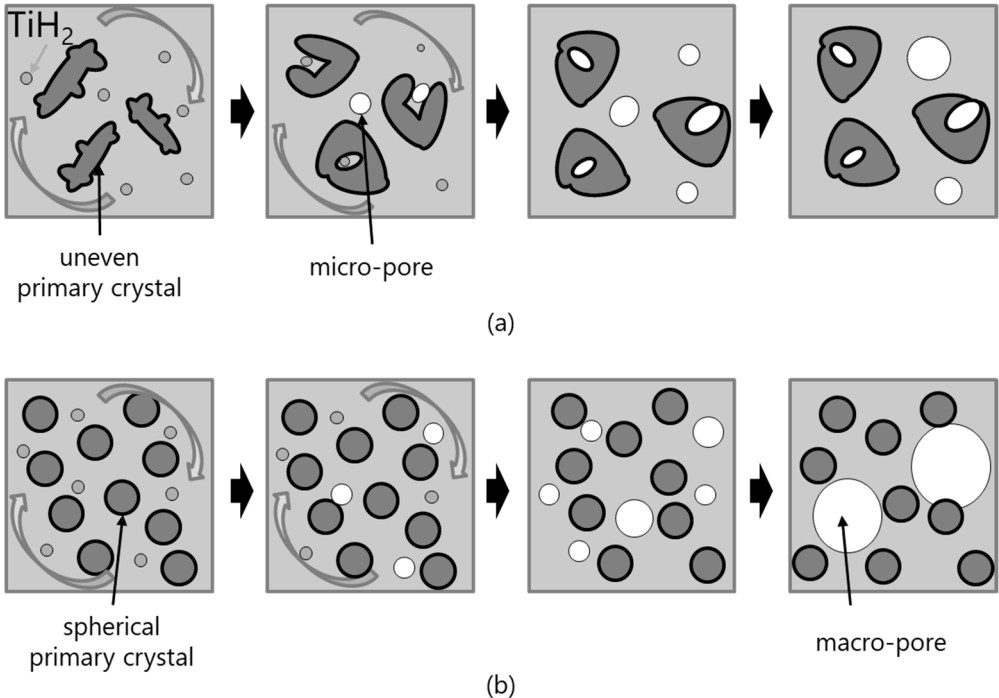

**Figure 13.** Schematic of the mechanism behind micro-pore formation in the (**a**) direct chill cast (DC-cast) and the (**b**) Strain-Induced Melt-Activated (SIMA) and slope-cast alloys.

## 4. Conclusions

A2024 alloy foams were fabricated in a semi-solid state using three different preparation methods for the master alloys. The most important finding of this study is that the morphology of the primary crystals influences the quantity of micro-pores inside the cell walls, but not on the quantity of macro-pores. The results can be summarized as follows.

1.　In the DC-cast alloy, coarse, dendritic, and uneven primary crystals were present before stirring. They were subsequently bent and spheroidized via stirring. On the other hand, the primary crystals in the SIMA and slope-cast alloys maintained their fine and spherical shape before and after stirring.

2.  The differences between the uneven primary crystals and the fine/spherical primary crystals before stirring had little effect on macro-pore morphology. The size of the primary crystals was significantly smaller than that of the macro-pores.

3.  In the DC-cast alloy, micro-pores existed inside the primary crystals after foaming. On the other hand, micro-pores were present between the primary crystals in the SIMA and slope-cast alloys. The number of micro-pores in the SIMA and slope-cast alloys was 1/3 of that in the DC-cast alloy.

**Author Contributions:** Writing–reviewing and editing, T.K. and S.S.; conceptualization, M.S. and S.S.; methodology, M.S. and S.S.; validation, M.S. and S.S.; formal analysis, T.K., M.S., and S.S.; investigation, M.S.; data curation, M.S. and S.S.; writing–original draft preparation, T.O.; visualization, M.S.; supervision, S.S.; project administration, S.S.

**Funding:** This study was supported by the Grant-in-Aid program of The Light Metal Educational Foundation.

**Acknowledgments:** The authors thank N. Sakaguchi from UACJ Corporation for suppling the A2024 ingots used in this study.

**Conflicts of Interest:** The authors declare no conflict of interest.

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
