# Peer review of "Effect of Primary Crystals on Pore Morphology during Semi-Solid Foaming of A2024 Alloys"

_metals, doi:10.3390/met9010088_

Round 1
Reviewer 1 Report
The authors produce A2024 foams by the direct-chill-cast, SIMA and slope casting method and show the differences in their structure. There problem is, there is no big innovation in this paper, there is also no clear aim but trying something and see what happens. I don´t see the scientific merit just by showing the structure and not discussing why there are some differences. What can I see or learn from Fig. 9? There is only one sentence describing Figure 9, 10a and 10b, and certainly no discussion about them. I total I just found some discussion from line 177 to 185, which is definitely not enough. Figures with less information should be omitted. English has to get a sever check. Some small spelling mistakes show, that the work was not done very accurately. References are very focused on the Asian region and miss to give a global overview of the topic
In detail:
L18: What is “the formation”?
L23: “was” not “were”
L36: “structure” instead of “pore formation”
L45: “as” instead of “an”
L86: “K/min” instead of “L/min”
L109: “…after processing by image the processing software …”. What does this mean?
L133: “All layers were porous, and did not have pores below the samples (Figure 5)”. This sentence has no sense.
L155: To define micro-pores as pores with diameters of less than 1000 microns may be formally correct, but there are some more logical definitions
Author Response
Thank you very much for your very careful reviews. We are sorry to have confused you in several ways. We reply to the suggestions. Please see the attachment.

Reviewer 2 Report
The authors have presented an experimental work where they study the pores in aluminum foams. The foams are produced from materials that were prepared with three different methods. The paper is interesting and with applications in modern industry. However, the paper needs to be revised before it can be accepted for publication.
First of all, the authors need to proofread their manuscript as it contains several grammatical errors or there are sentences with missing information, e.g. in page 6 "...most primary crystals had a of were less than 300 μm...".
Furthermore, the authors have a reference list that contains a few and rather old papers related to the topic. The authors need to enhance their reference list and exhibit the significance and novelty of their work.
It would be useful to note how many samples were measured. Additionally, how many measurements took place on the samples. Error bars are used in the graphs but they are not clear and require better figures and more explanation. Please elaborate more on the high error values, their explanation and the impact on the results.
Finally, refer to the most important findings, in the conclusions, quantitively.
Author Response

(The authors gave the same response as above.)

Reviewer 3 Report
In line 80 it can be seen that the temperature is very important for the fraction of solid. Therefore a few words should be written about the accuracy of melt temperature (+/-1 K?).
In line 86 the solidification rate cannot be given in L/min, probably it should be 3 K/min.
The sentence in line 140/141 is confused.
Author Response

(The authors gave the same response as above.)

Round 2
Reviewer 1 Report
The manuscript got a very substantial improvement, congratullations. I still do not agree with the definition of microporosity: If you will produce 50 mm pores and have some 5 mm pores in the cell wall, you can not define them as micropores. You could talk about intercellular porosity or similar. I found some definitions of microporosity in the literature for powder metallurgical foams, they talk of about 350 microns: Mukherjee M, García-Moreno F, Jiménez C, Rack A, Banhart J. Microporosity in aluminium foams. Acta Mater. 131, 156-168 (2017).
Author Response
Thank you very much for your very careful reviews. We reply to the suggestions. Please see the attachment.

Reviewer 2 Report
The authors have presented a new version of their paper which is improved in language. However, the paper has several drawbacks, mainly considering the number of specimens and the number of measurements. Furthermore, still the references included are not sufficient for proving novelty and scientific rigor of the paper.
Author Response

(The authors gave the same response as above.)
